# Hepatitis B and Liver Cancer: Community Awareness, Knowledge and Beliefs of Middle Eastern Migrants in Sydney, Australia

**DOI:** 10.3390/ijerph18168534

**Published:** 2021-08-12

**Authors:** Monica C. Robotin, Jack Wallace, Gisselle Gallego, Jacob George

**Affiliations:** 1School of Medicine, The University of Notre Dame, Sydney, NSW 2007, Australia; monica.robotin@nd.edu.au; 2Burnet Institute, Melbourne, VIC 3004, Australia; jack.wallace@burnet.edu.au; 3Centre for Social Research in Health, UNSW, Sydney, NSW 2052, Australia; 4Australian Research Centre in Sex, Health and Society (ARCSHS), La Trobe University, Bundoora, VIC 3086, Australia; 5Auburn Clinical School, School of Medicine, The University of Notre Dame, Sydney, NSW 2007, Australia; 6Storr Liver Centre, Westmead Institute for Medical Research, Westmead Hospital, Sydney, NSW 2145, Australia; jacob.george@sydney.edu.au; 7Westmead Clinical School, The University of Sydney, Sydney, NSW 2006, Australia

**Keywords:** hepatitis B, liver cancer, qualitative research, awareness, knowledge

## Abstract

Chronic hepatitis B (CHB) is a significant global health challenge given an increasing morbidity and inadequate public health response, Migrant populations are primarily affected by CHB in industrialised countries, and while more than 7% of Australians with CHB were born in Africa or the Middle East, little is known of their awareness or knowledge of viral hepatitis and its impact. This qualitative study, using semi-structured interviews with Assyrian and Arabic community leaders and focus groups (FG) with 66 community members sought to identify hepatitis and liver cancer knowledge and awareness among local Arabic and Assyrian-speaking communities in Western Sydney. Interviews were thematically analysed, with findings framing the topics for the FGs which were analysed using a framework analysis. Themes identified across both methods included limited awareness or knowledge of viral hepatitis or liver cancer, stigma associated with both conditions, variable levels of health literacy and trust in medical practitioners, and fear that receiving “bad news” would deter people from seeking care. Preferred sources of health information were family doctors, family members, the internet and the ethnic media. The study gave valuable information for the design of an educational program and provided useful information for the planning of culturally appropriate hepatitis screening and treatment services for these communities.

## 1. Introduction

In the space of a generation, the incidence of and mortality from primary liver cancer in Australia increased significantly and from 1982 to 2020 liver cancer moved from the 15th to the 6th most common cause of cancer death [1], with approximately 80% of liver cancers attributable to chronic viral hepatitis, which affects more than 400,000 Australians [2]. A large proportion of people with chronic hepatitis B (CHB) remain unaware of their diagnosis: of the 227,000 estimated to be living with CHB in Australia in 2018 [3], only 68% were diagnosed and just 9.3% were receiving antiviral treatment [2]. Approximately 56% of Australians with CHB were born overseas, with the largest population groups comprising people born in the Asia Pacific region (38%), Europe (10%) and Africa or the Middle East (7%) [3].

People born in hepatitis B endemic countries carry a disproportionate burden of CHB [3] and liver cancer [4], and understanding community awareness, knowledge and beliefs of these conditions in high risk communities is needed to improve effective clinical management. While a meta-analysis found a CHB seroprevalence of 2% across Middle Eastern and North African countries [5], rates ranged from 2–18.5%, encompassing low endemicity (e.g., Iran, Israel, Kuwait), as well as intermediate or high endemicity countries (including Iraq, United Arab Emirates, Egypt, Jordan, Yemen and Saudi Arabia) [6]. As the pooled estimates of CHB seroprevalence and odds of being chronically infected are significantly higher in refugees [5] and internally displaced persons [7], the burden of disease in the region may be actually significantly higher than currently estimated. 

A substantial body of research exists describing hepatitis knowledge and related beliefs of Asian immigrants to North America [8,9,10,11,12,13,14,15,16] and Australia [17,18,19,20,21,22], and has informed the design and implementation of hepatitis education, screening and clinical management programs and interventions in these communities [23,24,25,26,27]. Few Australian studies have investigated beliefs about and knowledge of hepatitis among people born in Middle Eastern and North African countries and their communities [28], although more recent work has investigated knowledge, beliefs and attitudes regarding viral hepatitis and liver cancer among people born in South Sudan living in Australia [29]. 

This current research was based in the Western and South-Western regions of Sydney, home to the largest number of people with CHB in Australia [30] and the only location in Australia projected to reach the Third National Hepatitis B Strategy treatment uptake target by 2022 [31]. In this region, approximately three quarters of the local population was born overseas, including about half of all Australians born in Iraq and more than a quarter of the country’s Syrian and Lebanese-born people [30]. Arabic is spoken at home by approximately 160,000 local residents, while about 50,000 (mostly Iraqi Christians) speak Assyrian or Aramaic [32]. Our study explored the Arabic and Assyrian-speaking local communities’ perceptions and beliefs concerning hepatitis and liver cancer and identify community discourses about this illness.

## 2. Materials and Methods

The research was developed and conducted with health workers affiliated with the Health Promotion Service in Sydney South West Area Health Service (SSWAHS) in partnership with Arabic and Assyrian community and religious organizations. 

Qualitative data were collected in two phases: (1) in depth semi-structured interviews with Assyrian and Arabic community leaders and (2) focus group discussions (FG) with Arabic and Assyrian community members. The findings of the semi-structured interviews informed the focus group methods. 

In the first phase, interviews were held with 12 community leaders from the Arabic and Assyrian communities and included people working in the health sector and others recommended by informants using snowball sampling. Potential participants were identified through our existing community networks and those of the project partners. This included Multicultural Health workers from the Health Promotion Service of two local health areas and local community and migrant services. The semi-structured interview guide used in phase one was developed and piloted with community health workers (CWHs) and included prompts to identify salient social and cultural issues in their communities, commonly encountered health issues, community knowledge and awareness of hepatitis. Interviews were conducted in English by two members of the research team (JW and MR), with all participants providing written informed consent. Eleven interviews were conducted face to face, and one conducted by phone. Each interview lasted 20–45 min, was electronically recorded and transcribed verbatim.

Phase 2 of the research consisted of focus groups (FGs) held with 66 people self-identifying as aged ≥18 years and born (or had a parent who was born) in a country where Arabic or Assyrian are spoken. Participants were purposively recruited and included CHWs and participants recruited through community health and advocacy organizations, faith-based groups and the networks of community health workers and community leaders. FGs were conducted separately with older and younger community members. FGs had between 8–11 participants for each language group (involving Assyrian and Arabic-speaking participants respectively). A guide, based on the findings of phase 1, explored awareness of health issues affecting the community, community awareness and understanding of hepatitis and liver cancer, enablers and barriers to hepatitis B testing. A trained bilingual community worker facilitated each focus group, which was attended by MR, who sat in the corner of the room with an interpreter, to follow the discussions and identify issues that may have required clarification. All participants provided in-language written consent for participation and for digital recordings and completed a brief demographic questionnaire. All FG were transcribed, with transcripts translated into English by certified translators. 

Two data analysis methods were used. First, a deductive thematic analysis of key informant data was conducted, with transcripts organised and managed using NVivo version 10 qualitative data analysis software (QSR International, Doncaster, Australia, 2012), with the findings guiding the development of topics to be discussed within the focus group. A framework analysis was used to analyse focus group discussions [33], with data summarized within a matrix developed in Excel, to facilitate pattern recognition. Both processes of analysis followed data familiarization, reading the transcripts, developing a coding framework (or matrix), coding transcripts and writing up of the results for discussion with the team. Findings of the community leader interviews were compared with the findings of the focus group analysis to achieve data triangulation and derive a broader perspective of the findings. Emerging themes were compared and contrasted and illustrative comments from participants selected to highlight relevant themes [34,35,36]. 

The study was approved by Human Research Ethics Committee of South West and Western Sydney Area Health Services (HREC/08/RPAH/47).

## 3. Results

### 3.1. Demographic Characteristics 

The 12 community leaders participating in semi-structured interviews (six Arabic and 6 Assyrian) were seven males and five females and included four medical practitioners who were members of these communities (two General Practitioners (GPs), one hepatologist and one pediatrician), two community workers/educators, two leaders of community and advocacy organizations, two teachers, a lawyer and a member of the clergy.

A total of 66 community members participated in seven focus groups each consisting of eight to twelve participants. Four FG were conducted in Arabic and involved 38 participants, of whom 58% were male with a median age range from 56 to 64, with countries of birth included Egypt, Lebanon, Syria, Palestine, Kuwait, Iran and Iraq. Three FG were conducted with Assyrian identifying people and involved a total of 28 people, with the majority (82%) being female. One FG comprised older Assyrian–speaking participants (median age of 71) and two were conducted in English with younger Assyrian participants (median age of 22 and 25 respectively). All participants in the older group were born in Iraq, as were 50% and 70% respectively of the younger group; the remainder were born in Australia. The demographic characteristics of FG participants are summarized in Table 1.

### 3.2. Themes

Common themes identified across all data included low awareness and knowledge of hepatitis and liver cancer, stigma associated with hepatitis and cancer, identification of barriers in accessing health care, and sources of health information. 

#### 3.2.1. Theme 1. Awareness and Knowledge of Hepatitis and Liver Cancer

High blood pressure, high cholesterol, diabetes, cancer and mental health issues were identified by community leaders and within the FGs as salient health issues within the community. Community leaders working in the health sector acknowledged the limited awareness and knowledge of hepatitis within Arabic and Assyrian communities with one GP comparing the experience of their community with that of people born in Asian countries. 

“*Asians, usually, have a bit of knowledge about it. Middle Easterners, not really much of it. And they don’t actually understand. You get the sense that they have no understanding. You do the education, but they have no idea of what it is actually about.*” (Male KI#1).

Within the FG, participants reflected on their limited knowledge of hepatitis transmission, and symptoms or availability of treatment. A poll conducted by the FG1 moderator established that 7 out of 11 participants had never heard of hepatitis with one participant noting that they would be unlikely to know about the infection unless viral hepatitis had specifically affected their lives:
“*Unless we have somebody that is infected, close or a family member, otherwise we never just naturally try to learn about disease if we are not involved with it*.” (Female FG1)

Several FG participants had heard of viral hepatitis, but this awareness was limited:
“*I know there is A, B and C, but I don’t know which is which and what the difference is*.” (Male FG1)

While some FG participants mentioned that hepatitis can be transmitted through infected blood and bodily fluids through tattooing, dental work, sharing razors, sexual intercourse and from mother to child, others thought that infection could occur by using dirty toilets, drinking infected water, or sharing food or water bottles with an infected person:
“*There are some food(s) that may contain the virus, especially take away food. They can get it from being unclean*.” (Female FG2).

Some participants voiced the belief that lifestyle factors may cause viral hepatitis, with an additional moral judgment being made as a result:
“*They don’t know what causes it (hepatitis) exactly. So, they always think it’s either cigarettes or alcohol, or they’ll ask you how you got it, then judge you basically*.” (Female FG3).

As has been reported in South Sudanese communities [29], hepatitis was thought to have occurred as a result of a big shock, or fear:
“*My husband had an accident at work in the oil company. He had a snake running all over him. Maybe through fear he got hepatitis.*” (Female, FG2).

Viral hepatitis infection is mostly asymptomatic. Some participants thought that any serious medical condition had to be symptomatic, and reported that within the community the experience of symptoms would be the motivator for seeking medical support, with a further acknowledgement that this would be further delayed by older members of the community:
“*How can something be wrong or missing in the body and you don’t feel pain. You have to feel pain*.” [Assyrian female, FG2].
“*Oh, I feel tired. Why would I get to doctor for that, or I look a bit pale, like I mean I’m not eating enough red meat, but you’re not going to go (to see the doctor).*“ (Assyrian male, FG4).

The experience in the participant’s country of origin in relation to expectations of medical professions were limited to the provision of services for people at the end of their life rather than any role related to disease prevention.

“*Back in the mother country you really only went to the doctor toward the end of life if there was something very serious. So that mentality of fear is hard to change. If you go to the doctor here, you find out you can have all these issues, which were never heard of back in the mother country*.” (Assyrian male KI#11).

While there was limited understanding of viral hepatitis, one symptom participants regularly associated with hepatitis was that of jaundice and for several there was an understanding that this was related to the liver, and in one case was the precursor to a devastating diagnosis.

“*We look at things in a very simple way. If we see someone is yellow, we say that they have something wrong with their liver*.” (Female FG2).

“*They retain water. My sister-in-law noticed that her stomach kept getting bigger and bigger...It ended up being fluid in the liver, cancer, she was all ruined.*” (Female FG2).

While awareness of hepatitis and how it could be treated was limited, some participants noted supportive care measures, including bed rest and a good diet and knew of foods (e.g., dates, honey, chicken liver and milk) which help recovery from liver ailments:
“*Before, little children would get liver infections and they would tell us to give them honey. The infection will go with honey*.” (Female FG2).

#### 3.2.2. Theme 2: Stigma Associated with Hepatitis and Liver Cancer

While there were mixed perspectives on whether there was a stigma related to infection with hepatitis or liver cancer, there was a clear sense of judgement related to the condition identified through the FG. The experience of liver disease was an issue largely to be kept within the family, with this separation informing the success of educational interventions targeting this community. One community leader noted a lack of effectiveness in one initiative which provided disease education to mothers as peer educators was unsuccessful, as:
“*Family members found it very hard to talk about it, sensitive, superstitious. Many family members were told not to talk about it, as we will now get bad luck and get the disease*.” (Arabic KI#12).

While there was little understanding of viral hepatitis, there were clear social, cultural and familial implications within these communities, particularly in the development of relationships:
“*In our mentality, we think that if anybody has hepatitis, he cannot get married. In his own mind, it seems like it is the end of my family, it ends here. It doesn’t just affect the person, it affects the whole family.*” (Female FG1).

Participants thought that the stigma related to hepatitis related to assumed modes of transmission such as through sexual transmission or through drug injecting:
“*There is always denial, stigma—strong stigma attached to it. I know people who had hepatitis because of their sexual relationships, and you know like that.*” (Assyrian KI#6).

This understanding and social implications relating to health generally, and specifically viral hepatitis, also affected the provision of health services, and the social marginalisation of health care providers within the community. One community leader who was a medical practitioner noted that that some patients avoided doctors in social situations so as to not to raise the suspicion that they are ill:
“*I’ve become known as the hepatitis doctor or the liver doctor, so some people will go out of their way not to talk to me.*” (Arabic KI#12).


People with viral hepatitis are at greater risk of developing liver cancer and other liver disease. Within these communities, cancer has a stigma with (especially older) people avoiding referring to it at all costs: “They do the [sign of the] cross before they even say it,” (Female FGD 4); or, if this is unavoidable, they use euphemisms and protective language [35], such as “the ugly disease”, or “that thing”. One participant referred to cancer in the following terms: “It’s like Voldemort, He who shall not be named.” (Female, FG4).

Younger participants, while less reticent to mention cancer, thought families preferred to keep cancer a secret for fear of collective judgement and because of a belief that cancer occurs as a retribution for past misdeeds:
“*They’re all concerned about what people will think of them, what people will say of them, and especially if they had made a mistake in their life. A friend or a colleague will remember that one thing that they had done in their past and they will blame that thing, they’ll say that God is punishing you for that*.” (Male FG4).

A cancer diagnosis can also be kept secret from one’s own family, and the family may not disclose the diagnosis even after someone dies from cancer:
“*There was a bride, already dead, and when her family was asked ‘What happened to her? She was only a young bride’, the answer was only ‘Oh, that disease, that disease’. They would still not want to tell people, even after the family member died, that she was suffering from cancer, because they regard it as a shame.*” (Arabic female, FG1).

The very mention of the work “cancer” needed to be avoided even within clinical settings as disclosing a cancer diagnosis to a patient may hasten their demise and should be avoided:
“*Again, I’m always going back to the culture thing, because in their upbringing, the medical system in Iraq wasn’t like you sit in front of the patient and tell them, “You’ve got cancer, you got ten months to live. They’d be like, “Okay, don’t tell –” you don’t tell the children, don’t tell that person that they’ve got a bit of time left.*” (Male FG3).

#### 3.2.3. Theme 3: Barriers to Accessing Health Care

Some participants identified low English proficiency, limited education, and health literacy as key barriers in accessing health care:
“*Those who have studies and are educated would know, but those who haven’t, wouldn’t have a clue about it.*”(Female FG2).

However, literacy and health literacy levels were variable between and across the Arabic and Assyrian communities with one community leader noting:
“*Prior to the war (Iraq had an) absolutely fantastic health system, good education system, good health literacy…. but still you are talking about people from Egypt for instance who are coming who are not that literate*.” (Arabic female KI#3).

According to younger participants, the elderly tend not to question what the doctor says and use younger family members as informal interpreters, to explain medical terminology and clarify what the doctor said. As not all information is translated or explained, patients often get an incomplete picture of their condition:
“*I’ve seen younger people going with their parents who doesn’t know the language and the parents saying to the kids, ‘What the doctor said?’ They shush them down. ‘Nothing, nothing,’ and I know what they’re talking about. ‘Nothing, nothing,’ you just need to take this, this and that, but there’s more information the parents need to know.*” (Male FG4).

#### 3.2.4. Theme 4: Sources of Health Information 

Generally, doctors are held in very high regard and people trust their GPs especially: *“a lot of Arabic speaking people are very trusting of their GP. If the GP says do these test, then they will do it.”* (Female Arabic KI#3). However, doctors are often pressed for time and do not give patients the opportunity to ask questions, or express concerns: “*Some doctors don’t even give you 5 minutes with them. They just ask you questions and want you to leave straight away. Even if the patient has any questions, they don’t have the time to ask them or forget them from how rushed they feel*.” (Female FG2).

While family doctors were identified as a preferred source of health information, family members with a lived experience of an illness also had currency in the provision of health information, and were often trusted more than doctors:
“*If someone has been through the same thing … I’ll just go up to her and be like, ‘Oh, what did you do to get through it?’ and they do it a lot. You just see what the others are doing, and you start doing it too just because maybe it’s working for them.*” (Female FG3).

One community leader noted the importance of educating other community leaders to improve knowledge within the broader Arabic and Assyrian communities:
“*If you want to make it a big issue, then you need to educate the rest of us [community leaders] …as a leader in the community, I need to know just a little bit more, just to convey the message and make it urgent*.” (Arabic Female KI#3).

Ethnic media was noted as a valuable source of information for people from these communities:
“*A lot of people listen to the TV in their own languages... If you go and put it in the media—go and check for your Hep B and Hep C, there is a treatment. Go and see a doctor, there will be a big response if it goes through the ethnic*.” (Arabic male KI#1).

Suggestions for providing health education included using community radio, the ethnic press and the Assyrian Resource Centre, as this can screen educational videos, preferable to written information which commonly “goes in the bin” (Assyrian female KI#6). Using a range of communication channels is advisable: “We just have to cover a whole range of different strategies when it comes to that.... You need the radio, you need the media or with the radio, or the print media, but this is to make it an urgent need out there.” (Arabic female KI#3).

Community meetings and Church gatherings could be used to raise awareness of viral hepatitis: “I can guarantee that our people, they don’t know about hepatitis but, if we can talk with them, or organise some meeting or conference about that, why not? Because we were a minority in Iraq, the Chaldean community, in a big Muslim community, we used to see the Church like everything for us, not fully a religion path. Also a social path, also an educational path.” (Assyrian male, KI#2). 

Several community leaders noted, however, that while many Assyrians speak Arabic, reliance on written hepatitis resources can pose problems as many Iraqis do not read and write Assyrian while some Arabic speakers may not be literate in their own language.

## 4. Discussion

Our study found low awareness and knowledge of viral hepatitis and liver cancer among people from Arabic and Assyrian speaking countries, with low awareness and knowledge about types of hepatitis, transmission and treatment. Low English proficiency and health literacy are significant barriers in accessing health care and a stigma is attached to both hepatitis and cancer. It is interesting to note that CHWs also had very little knowledge about hepatitis B. This is not surprising, as in Australia CHWs have different roles and serve a range of functions in various contexts. There are no definitions around the scope of CHW interventions, and the utilization of CHWs. Furthermore, there is no formal training, and they are not required to have formal qualifications or registration [37].

Our results corroborate the findings of other studies reporting limited understanding of viral hepatitis in high-risk communities with higher disease prevalence [11,36,38,39,40]. Like Sweeney et al., we found that viral hepatitis does not figure in the community discourse about illness [36], a fact also noted in a recent systematic review examining the health discourse of Arabic people living in the United States [41]. 

Participants were unaware of the link between hepatitis and liver cancer. Saleh et al. found that communication about cancer in Arab Australians is hampered by fear and disease stigma, with euphemisms and protective language (such as “that illness”, or “the bad sickness”) likely to be used [35]. Our participants used this approach to avoid referring to cancer and believed that non-disclosure of a cancer diagnosis to affected family members was prevalent. While Arab-Australians submit to “God’s power” to bring about cancer, individuals also have a responsibility to do their best in fighting the disease and getting adequate treatment [35].

Despite iterative searches of the English literature, we found few studies (both quantitative and qualitative) on the knowledge, attitudes and practice around hepatitis B of people originating in Arab countries, with available studies suggesting significant knowledge gaps. The level of knowledge and awareness among patients diagnosed with hepatitis C in Egypt (the country with the globally highest hepatitis C prevalence) was low regarding disease transmission, beliefs about vaccine availability and treatment availability [42]. Similar findings were noted among internet and social media users in Saudi Arabia; while people with a higher level of education or working in the medical field were more knowledgeable, and fewer than a third had satisfactory knowledge of disease transmission, treatment and vaccination [43]. Two thirds of a large sample of Iraqi inpatients (over 17,000 participants) had low knowledge of hepatitis B and hepatitis C: the majority had heard of a disease called ‘hepatitis’, but only about two thirds and a third respectively had heard of hepatitis B or hepatitis C specifically and fewer than 20% had previously had been tested for hepatitis [44].

Some beliefs about disease transmission seem to transcend cultures and regions: the belief that hepatitis can be transmitted through sharing food or cooking implements appears prevalent in groups with high hepatitis prevalence, such as Chinese migrants [45]. The belief that hepatitis can be caused by a big shock, or fear is also shared across cultures: Traditional Chinese Medicine views the liver within a system of correspondences encompassing the body and external phenomena believes that negative emotions (such as anger, or anxiety) can be potential causes hepatitis [45].

Our study used complementary collection methods and sought both the perspectives of community leaders and community members. The triangulation achieved through using both semi-structured interviews and FG strengthens the quality of our interpretive findings [46]. This confirms findings of our knowledge survey of 270 Arabic and Assyrian community members with a relatively low hepatitis knowledge score (6.2 ± 2.9 out of 11), with the lowest scoring questions being those about whether hepatitis B can be transmitted through food and drink, if infected people can look and feel healthy, whether hepatitis B is a life-long illness and whether it can cause liver cancer. 

## 5. Conclusions

A concerted effort is needed to increase awareness of hepatitis and liver cancer in Middle Eastern Australians, involving educational campaigns using a range of communication methods. To our knowledge, our study is the first to assess attitudes and practices around hepatitis in Australians from a Middle Eastern background, augmenting the limited body of knowledge of the topic. The findings of a low-level of knowledge and understanding of hepatitis and liver cancer in these groups suggests that this needs to be factored in to effectively address screening barriers, enhance disease awareness and promote treatment uptake [45,47].

## Figures and Tables

**Table 1 ijerph-18-08534-t001:** Socio-demographic characteristics of FG participants.

	FGD1 Older Arabic (%)	FGD2 Older Assyrians (%)	FGD3 Young Assyrians (%)	FGD4 Young Assyrians (%)	FGD5 Arabic (%)	FGD6 Arabic (%)	FGD7 Arabic (%)	TOTAL N = 66 (%)
**No of participants**	11	12	8	8	8	9	10	66
**Gender**								
Male	3	0	2	3	5	4	10	27 (41)
Female	8	12	6	5	3	5		39 (59)
**Median age** (years)	56	71	22	25	64	59	60	51
**Time lived in Australia**								
<1 year	-	-	-	-	-	1 (11)	-	1 (2)
1–5 years	3 (27)	3 (25)	-	-	-	2 (22)	4 (40)	12 (20)
5–10 years	2 (18)	3 (25)	2 (25)	-	1 (12.5)	3 (33)	-	11 (18)
10–20 years	1 (9)	5 (42)	3 (37.5)	6 (75)	3 (37.5)	2 (22)	1 (10)	21 (34)
>20 years	5 (45.5)	1 (8)	3 (37.5)	2 (25)	4 (50)	1 (11)	-	16 (26)
**Education level**								66
Primary school	-	7 (58)	-	-	4 (50)	5 (56)	4 (40)	20 (30)
High school	6 (55)	3 (25)	2 (25)	2 (25)	3 (38)	2 (22)	1 (10)	19 (29)
Tertiary	5 (45)	2 (17)	6 (75)	6 (75)	1 (12)	2 (22)	5 (50)	27 (41)
**Languages spoken at home ***								
Arabic	11	1	-	-	8	9	8	36
Assyrian	-	12	8	8	-	-	1	29
English	1	-	3	6	1	2	1	14

* Some respondents spoke more than one language at home.

## Data Availability

Not applicable.

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
