# Peer review of "Hepatitis B and Liver Cancer: Community Awareness, Knowledge and Beliefs of Middle Eastern Migrants in Sydney, Australia"

_ijerph, 2021, doi:10.3390/ijerph18168534_

Round 1

Reviewer 1 Report

Dear sir, thank you to select me to review an article Robotin MC et al. Hepatitis B and Liver Cancer: Community awareness,  knowledge and beliefs of Middle Eastern migrants in Sydney, Australia. Topic of the manuscript is interseting, paper is well written. The authors concluded that Asyrian and Arabic community have a low-level knowledge and understanding of hepatitis and liver cancer.

I recommend one change: 

Please describe in detail principles of selection of community leader and construction of focus groups.

My evaluation is minor revision.

Author Response

A description has been added on how community leaders were selected and more information on the focus groups.   

Reviewer 2 Report

  1. Abstract: Is this a good expression? ‘with findings framing the methods used for the FGs’

The FG is the method. I would say it should be: ‘ with findings framing the topics for the FGs’. Please consider.

  1. Last sentence of the abstract. Is educational resource a good expression? Educational program seems better to me. Is the educational resource or program already available? If no, please reconsider the sentence. The study gave valuable information for the design of an educational program, would that be correct? Please consider.
  2. The focus group participants were recruited from community health workers and still they had very little knowledge about Hepatitis B. I think this is surprising and should be discussed in the discussion section. What kind of health workers participated in the FGs. Maybe this could be explained in more detail.
  3. It is unusual to have references in the conclusion. The conclusion comes from your study. Please consider.

Thank you for the chance to read this interesting manuscript.

Author Response

Please find attached the responses to reviewer No. 2
